# A Bittersweet Computational Journey among Glycosaminoglycans

**DOI:** 10.3390/biom11050739

**Published:** 2021-05-15

**Authors:** Giulia Paiardi, Maria Milanesi, Rebecca C. Wade, Pasqualina D’Ursi, Marco Rusnati

**Affiliations:** 1Experimental Oncology and Immunology, Department of Molecular and Translational Medicine, University of Brescia, 25123 Brescia, Italy; g.paiardi@unibs.it (G.P.); m.milanesi006@unibs.it (M.M.); 2Molecular and Cellular Modeling Group, Heidelberg Institute for Theoretical Studies, Schloss-Wolfsbrunnenweg 35, 69118 Heidelberg, Germany; rebecca.wade@h-its.org; 3Center for Molecular Biology (ZMBH), DKFZ-ZMBH Alliance, Heidelberg University, Im Neuenheimer Feld 282, 69120 Heidelberg, Germany; 4Interdisciplinary Center for Scientific Computing (IWR), Heidelberg University, Im Neuenheimer Feld 205, 69120 Heidelberg, Germany; 5Institute for Biomedical Technologies, National Research Council (ITB-CNR), 20054 Segrate, Italy

**Keywords:** molecular modeling, molecular docking, molecular dynamic simulations, glycosaminoglycans, heparin, heparan sulfate

## Abstract

Glycosaminoglycans (GAGs) are linear polysaccharides. In proteoglycans (PGs), they are attached to a core protein. GAGs and PGs can be found as free molecules, associated with the extracellular matrix or expressed on the cell membrane. They play a role in the regulation of a wide array of physiological and pathological processes by binding to different proteins, thus modulating their structure and function, and their concentration and availability in the microenvironment. Unfortunately, the enormous structural diversity of GAGs/PGs has hampered the development of dedicated analytical technologies and experimental models. Similarly, computational approaches (in particular, molecular modeling, docking and dynamics simulations) have not been fully exploited in glycobiology, despite their potential to demystify the complexity of GAGs/PGs at a structural and functional level. Here, we review the state-of-the art of computational approaches to studying GAGs/PGs with the aim of pointing out the “bitter” and “sweet” aspects of this field of research. Furthermore, we attempt to bridge the gap between bioinformatics and glycobiology, which have so far been kept apart by conceptual and technical differences. For this purpose, we provide computational scientists and glycobiologists with the fundamentals of these two fields of research, with the aim of creating opportunities for their combined exploitation, and thereby contributing to a substantial improvement in scientific knowledge.

## 1. Introduction

In 1902, Hermann Emil Fischer, a German chemistry professor, was awarded the Nobel Prize in Chemistry for his studies on sugar and purine synthesis. Since then, many other scientists have been awarded with the Nobel Prize for glycobiology-oriented studies, including Karl Landsteiner in 1930 for the discovery of human blood groups and Luis F. Leloir in 1970 for the characterization of carbohydrate biosynthesis. Currently, the number of glycobiology-oriented studies is exponentially increasing, showing that sugars are being found to be involved in a growing number of physiological and pathological processes.

Among the various classes of sugars, glycosaminoglycans (GAGs) are linear polysaccharides that can attach to core proteins to form proteoglycans (PGs). GAGs and PGs are widely distributed in the bodily fluids, and can be found to be associated with the extracellular matrix (ECM) or expressed on the cell membrane. They are endowed with a mind-boggling diversity of structures, providing a high level of variety and specificity to a wide array of biological functions. Considering the huge amount of data on the functional involvement of GAGs/PGs in physiological and pathological processes, relatively little progress has been made towards truly understanding the molecular mechanism(s) by which GAGs/PGs bind and “tweak” proteins. This is possibly due to the complexity of the structure of GAGs/PGs that has so far prohibited the development of appropriate analytical technologies and experimental models for their study.

This problem is well exemplified by considering the “omics” branch of science (genomics, proteomics, lipidomics, glycomics and interactomics) aimed at characterizing and quantifying large pools of biomolecules and their interactions, and at translating this information into structures, functions and dynamics. Over the last 30 years, glycomics has not been able to keep up with the rapid progress in genomics and proteomics. Only recently have we witnessed significant advances in new and powerful omics methods that have improved our knowledge of glycomics [1,2,3], and of “glycosaminoglycanomics” and “proteoglycomics” in particular [4,5,6].

Among the computational methods that can boost the understanding of how GAGs/PGs bind to proteins, particularly promising are molecular modeling, docking and molecular dynamics (MD) simulations. In effect, by working in a virtual environment, these methodologies benefit from a high resilience and potential for high throughput [7,8,9,10]. Briefly, molecular modeling uses molecular mechanics models to construct three-dimensional molecular structures; molecular docking gives favorable arrangements of molecules in complexes (e.g., GAG/protein complexes); MD simulations reproduce the dynamic behavior of individual molecules or complexes. Put in simple terms, the relationship between molecular modeling and MD simulations is similar to that existing between photography and cinematography: the former describes the structure of a molecular system, usually at an atomic detail level, in a “static” way. The latter instead allows the description of the dynamic behavior of a molecular system through the solution of Newton’s equations of motion using the classical laws of physics. In this way, MD simulation acts as a “computational microscope” that provides a “real-time visualization” of phenomena such as peptide folding, protein conformational changes and protein–protein interactions considering the flexibility of the molecules and the possible conformational changes induced by mutations or by the perturbation of the environment (e.g., modification of the pH or of the salt concentration [11,12]).

Many reviews have been published on GAGs/PGs [13,14,15,16] and on the latest developments in computational studies on GAGs/PGs [7,8,9]. In this review, we attempt to bridge these two fields of research that have so far been kept apart by conceptual and technical differences, meaning that computational approaches have not yet been fully exploited for studying GAGs/PGs. We aim to provide computational biologists and glycobiologists with the fundamentals of the two different fields of research, while emphasizing the opportunities for computational approaches to the study of GAGs/PGs.

## 2. Fundamentals of GAGs and PGs

The first reported study on a GAG dates back more than 80 years [17]. However, much remains to be learnt, especially from “omics” approaches that have become mandatory for a comprehensive understanding of the structure/function relationships of biological macromolecules.

### 2.1. Structure, Biosynthesis and Distribution of GAGs

GAGs are highly heterogeneous, negatively charged polysaccharides. Different combinations of different hexuronic acids and amino sugars result in five main classes of GAGs, distinguishable by the composition of their disaccharide units [13,18]: hyaluronic acid (HA) [19], chondroitin sulfate (CS) [20], dermatan sulfate (DS) [21], keratan sulfate (KS) [22], and heparan sulfate (HS)/heparin [14] (Figure 1A).

HA is assembled at the plasma membrane, is not linked to core proteins, and remains unsulfated [18]. In contrast, the biosynthesis of all the other GAGs occurs at the Golgi apparatus where they undergo a sequential process consisting in linking to a core protein (to form PGs), chain elongation (mainly catalyzed by glycosyltransferases encoded by the tumor suppressor EXT family genes [23]) and finally, chain modifications (mainly catalyzed by sulfotransferases [24], which introduce sulfated groups in the disaccharide units of all the GAGs except HA) (Figure 1A).

CS, DS and HS contain a common tetrasaccharide (4-mer) linker that is O-linked to specific serine residues in core proteins. KS can instead have three different linkers, either N-linked to asparagine or O-linked to serine/threonine residues in core proteins [15]. Multiple linkers can be attached to a core protein. Then, GAGs are elongated, leading to the synthesis of chains composed of 10–200 repeating disaccharide units linked by glycosidic bonds. Importantly, the core protein of a PG is synthesized in a template-driven manner, but its GAG chains are subsequently added in a non-template-driven synthetic process, thus contributing to the broad heterogeneity of the GAG chains composition (Table 1).

The process of sulfation of GAGs is of importance to determine their structural heterogeneity and interaction potential (further discussed in Section 2.2). It has been extensively studied for heparin and HS, where 2-O- and 6-O-sulfation occur only after C5 epimerization (that in turn requires prior N-deacetylation/N-sulfation). Consequently, the distribution of 2-O- and 6-O-sulfate groups is restricted to N-sulfate regions [25]. The modification process in heparin is more complete than in HS. As a result, the heparin structure is more homogeneously composed of regular trisulfated disaccharide sequences made up of alternating, α-1,4-linked residues of iduronic acid (Ido)A2S and N,6-disulfate D-glucosamine (GlcN). These regular sequences are occasionally interrupted by nonsulfated uronic acids (either glucuronic (GlcA) or IdoA) and by undersulfated hexosamines (GlcNS, GlcNAc, GlcNAc6S). The less extensive modifications that occur during the biosynthesis of HS lead to GAG chains characterized by a low IdoA content, low overall degree of O-sulfation and a heterogeneous distribution of the sulfate groups. Eventually, disaccharides containing GlcNAc or GlcNS may form clusters ranging from 2 to 20 adjacent GlcNAc-containing disaccharides and 2–10 adjacent GlcNS-containing disaccharides. However, about 20–30% of the chains contain alternate GlcNAc and GlcNS disaccharide units [26]. As these modifications are incomplete in vivo, not all of the sugar residues are modified, thus contributing to the structural heterogeneity of GAGs (Table 1).

Once assembled, PGs can remain segregated into intracellular granules, or become exposed on the plasma membrane, secreted in body fluids or deposited in the ECM (Figure 1B). Interestingly, besides their direct synthesis, the free forms of GAGs and PGs can result from cleavage of the polysaccharide chains or of the core protein of PGs, respectively, by glycosidases or proteases [27,28], further adding to the structural and, hence, functional complexity of GAGs/PGs (Table 1).

PGs are divided into four major classes, depending on their extracellular and intracellular localization. The only intracellular PG is serglycin, which carries heparin as the polysaccharide chain and is segregated in the granules of mast cells. Importantly, the heparin chains of serglycin can be depolymerized by endoglycosidases to obtain free heparin that is then released to mediate a long list of biological activities [29]. At the cell surface and in ECMs, the most represented PGs are those carrying HS chains (heparan sulfate proteoglycans, HSPGs). They can associate with the cell membrane at concentrations of 10^5^–10^6^ molecules/cell either via a transmembrane core protein or via a glycosyl-phosphatidyl-inositol (GPI) anchor. Syndecans are the most represented family of transmembrane HSPGs [30], and their cytoplasmic domain can interact with the cytoskeleton and can transduce a signal inside the cell upon binding with their extracellular ligands [31]. Glypicans are instead GPI-anchored HSPGs whose main function is to facilitate and/or stabilize the interaction of different cytokines and growth factors with their receptors and to transport cargoes into and through cells for their recycling [32]. Perlecan and agrin are the two most prevalent PGs in the basement membranes, but they can be also found at the cell surface, anchored to integrins or other receptors [16]. Extracellular PGs represent the largest PG family. This family includes small leucine-rich PGs (SLRPs) and hyalectans (e.g., aggrecan and versican), key structural components of cartilage, blood vessels and the central nervous system, which bind HA and thereby form supramolecular complexes of high viscosity [16].

The composition of GAG/PGs changes during cell differentiation [33], and their expression profile can be significantly different among differentiated cell types [34]. Moreover, the length, sequence, sulfation degree, membrane association, extracellular shedding, and levels of expression of GAGs/PGs themselves and of glycosidases undergo pronounced modifications in pathological conditions such as inflammation [35] or cancer [36,37], with some PGs even being used as markers for prognosis [38]. All these modifications further add to the structural and functional heterogeneity of GAGs/PGs (Table 1).

### 2.2. Biological Functions of GAGs and PGs

Despite the great variability of their structures and distribution in nature, GAGs/PGs share a high interaction potential, both in terms of the type and the amount of ligands that they can bind. GAGs/PGs have been demonstrated to bind to each other (e.g., hyalectans and HA mentioned above) and lipids (as occurs in synovial joints to allow lubrication [39]). More importantly, they bind a wide array of proteins, including growth factors, cytokines, proteases, coagulation enzymes, and proteins of the ECM [28,40,41]. These interactions usually occur between the negatively charged groups present on the polysaccharide chain (either COO^−^ groups in HA or SO_3_^−^ groups in all the other sulfated GAGs) (Figure 1A) and stretches of cationic amino acid residues (mainly arginine and lysine) present in proteins and referred to as “basic domains” or “heparin-binding domains”. Basic domains can consist of either linear amino acid sequences or conformational domains formed by non-contiguous basic amino acid residues. Multiple basic domains can sometimes be found in the same protein, conferring a higher capacity to bind to GAGs. In general, GAG/protein binding is electrostatic in nature, with relatively low affinity (ranging from low μM to high nM) compared to specific ligand/receptor or antigen/antibody interactions (ranging from low nM to pM) [42,43].

In general, the long saccharide chains of GAGs/PGs allow multiple bindings with several copies of a protein, inducing effects such as the increase in protein concentration in the microenvironment and the protection from proteolysis and thermal degradation. Additionally, the multivalent binding of a protein to GAGs/PGs can induce its oligomerization [44] and/or allosteric effects [28], that, in turn, can facilitate the binding of the protein to its actual receptor (Figure 2).

By these mechanisms, GAGs/PGs exert functions that range from relatively simple mechanical support functions (mainly when present in the ECMs) to more intricate effects on cellular processes such as cell proliferation, differentiation, adhesion and migration (when associated with the plasma membrane), with consequences in different physiological processes, including development and tissue homeostasis. They are also involved in important pathological processes, such as tumor neovascularization, growth and metastasis, neurodegeneration and viral infection. Finally, GAG/PGs regulate inflammation and the immune responses [16,41,45,46]. On the basis of their involvement in pathological processes, GAGs/PGs have been considered as therapeutic targets or as templates for the development of heparin-like HSPGs-antagonists able to bind and sequester pathological proteins hampering their interaction with HSPGs co-receptors with therapeutic benefits [43,47,48].

In conclusion, the characterization of the chemical structures of GAGs/PGs and of their binding modes to protein partners is mandatory for the comprehension of biological processes involving GAGs/PGs. Furthermore, it is a necessary basis for the design of new drugs aided by molecular modeling, docking and MD simulations.

## 3. Fundamentals of Molecular Modeling, Docking and MD Simulations in Glycobiology

The term “molecular modeling” is commonly understood to comprise all the methods used to model and simulate the behavior of molecules in silico, including molecular docking and MD simulation. Here, we will consider a narrower definition and discuss the three methods separately.

### 3.1. Molecular Modeling of GAGs

The aim of the molecular modeling is to construct models of the three-dimensional structure(s) of molecule systems considering physico-chemical features, such as geometry, energy, and electrostatic potential. Such structures may be determined by experimental techniques including X-ray crystallography (Figure 3), nuclear magnetic resonance spectroscopy (NMR), cryogenic electron microscopy (Cryo-EM), small-angle X-ray scattering (SAXS), small-angle neutron scattering (SANS), quasielastic neutron scattering, dynamic light scattering, solution scattering, fiber diffraction, electron paramagnetic resonance and Förster resonance energy transfer and made freely available in data banks (Table 2).

Nevertheless, collecting such structural data for GAGs alone or complexed with proteins remains a challenging task, since GAGs tend to assume a wide distribution of conformational states that make them refractory to X-ray or cryo-EM crystallization. Moreover, NMR, which performs well in the case of flexible structures, has some limits when used to solve long structures such as GAG polysaccharidic chains alone or complexed to proteins.

As mentioned above, the lack of appropriate GAG structural data has delayed the exploitation of molecular modeling, docking and MD simulations in glycobiology. However, to compensate for the lack of GAG experimental structures, an increasing number of popular web-based tools with dedicated features for in silico modeling of glycans have been developed and released in the few last years (Table 2).

Although only recently released, web-based tools for in silico GAG modeling have quickly gained prominence with respect to experimental approaches (53% from computational modeling vs. 48% from experiments of all the GAG models reported in the literature since 1990, Figure 4).

Among the experimental methods, X-ray crystallography has been the prime method to solve the structures of short GAG/protein complexes, due to the stabilizing effect exerted by the protein on the GAG, which would otherwise be too flexible to be crystallized. The 12-mer heparin model obtained by NMR (PDB id 1HPN, Figure 3) is the main starting structure adopted for subsequent molecular docking and simulations of heparin [49].

Among the in silico molecular modeling software packages, AMBER-tleap, GAG-builder and MOE are the most frequently used. Significant is also the use of public databases (PubChem, Zinc, DrugBank, EMBL-EBI, KEGG, GAG-databases, monosaccharides databases) and in-house libraries, including FDA-approved drugs, pseudo-disaccharide libraries and LOPAC [80], that provide both 2D and computed 3D structures of GAGs. It must be pointed out that the molecular modeling of GAGs remains a time-consuming process that still requires tedious manual refinements [81]. Additionally, although these methods allow the modeling of long GAG chains [82], the study of their interaction with other biomolecules remains challenging.

### 3.2. Molecular Docking of GAGs with Their Targets

It goes without saying that the limitations described for molecular modeling of GAGs impact their molecular docking to ligands. Molecular docking computes the configuration of a ligand–receptor complex by calculating the most favorable arrangements. In molecular docking, each of the two molecules involved in the complex is described by its dihedral angles, bond lengths and bond angles, which define its geometry and overall structure [83]. Unfortunately, unlike some small ligands that interact with well-defined binding pockets in proteins, GAGs bind to large protein surfaces primarily through electrostatic interactions, making the calculation of the optimal arrangements very difficult. Due to their charged nature, consideration of electrostatic and water-mediated interactions is necessary to understand GAG binding modes. The main structural features of GAGs that pose difficulties for molecular docking studies are listed in Table 3.

Other obstacles are the flexibility of the whole GAG chain (depending upon the 1–4 glycosidic linkage between the monosaccharide units), that of the functional groups on the monosaccharides, and the structural instability of GAG binding sites on the protein partner that can undergo conformational changes (“induced fit”) upon interaction with the GAG. Thus, computational docking of GAGs to proteins remains extremely challenging [84].

The main docking software program used to compute the interaction of small GAGs with proteins is Autodock (Figure 5). Even though it was originally written to compute the interactions between macromolecules and small ligands, its parametrization is suitable for docking small GAGs to proteins. It is, however, limited by the number of free torsions that can be considered in the ligand (up to 32). This is an important limitation if we consider that a small 4-mer heparin contains 28 torsions. Such constraints have surely contributed to the fact that the majority of computational studies on GAGs has been performed with short saccharide chains (further discussed below). Besides Autodock, other docking programs used to compute GAG/protein interactions include Autodock-Vina, Glide, Dock, Gold and HADDOCK (Table 2). Other docking software programs specifically dedicated to sugars have been released recently (e.g., VinaCarb, Glycotorc-Vina and GAG-dock (Table 2 and Figure 5)).

Nevertheless, all that glitters is not gold. Indeed, even though the quality of the FFs by which GAGs are described has improved, the length and the number of free torsions of longer GAGs still impact the computation time, confining the predictions of GAG/target interactions to short saccharide chains. Notably, novel approaches have been proposed to overcome the “free torsion-limitation” issue, including an incremental docking method in which small GAGs are flexibly docked and connected following a pre-defined path and the final long-GAG/target complex is refined by MD simulation [44], an automated fragment-based approach in which trimeric GAGs are flexibly docked on a protein binding site assembled and refined by MD simulations [81], the use of mono/disaccharide probes to identify heparin-binding sites at which to perform local docking of longer GAGs by Autodock/DOCK [85] and the possibility to introduce solvent into the binding site prior to docking [86].

### 3.3. MD Simulations of GAGs and GAG/Target Complexes

The history of MD simulations started more than 60 years ago, when Alder and Wainwright carried out the first simulation of a phase transition in a system of hard spheres [87]. However, we needed to wait until 1977 for the first MD simulation of a protein [88] and until 1985 for that of heparin [89].

Then, slowly, the groundwork that made MD simulations a reliable process resulted in the 2013 Nobel prize being given to Karplus, Levitt and Warshel for the development of multiscale models for complex chemical systems [90]. From then on, MD simulations have gained popularity in glycobiology, due to the increased number of available structures in the PDB, to the release of software specifically dedicated to GAGs (Table 2), and to the implementation of computer technologies such as high-performance computing, that allow easier use of the techniques and decrease the computing time. Although the MD simulation of GAGs remains burdened (see Table 3), the interest in the field is increasing. A growing amount of work is mainly devoted to the comprehension of the dynamic behavior of GAGs (which are characterized by an ensemble of conformations rather than a single secondary or tertiary structure). Additionally, there is an increased focus on the characterization of the conformational changes occurring in GAGs and proteins following their mutual interaction [9].

The classical all-atom MD simulation method is mainly used to study GAGs/PGs and consists of numerically solving coupled equations of motion for a system in which the atoms move at defined velocities. The result of these calculations consists of a series of trajectories of the biomolecules, from which thermodynamic and dynamic properties of the system can be extracted. Importantly, the reliability of the prediction of the behavior of a system depends on the assumptions used to describe the interactions within it. Thus, the parameters chosen to describe the systems must be as realistic as possible, considering not only temperature and pressure, but also other relevant features, such as water models, pH and salt concentration of the solution, that are particularly relevant when working with GAGs [91]. The prediction of how atoms and molecules interact with each other in conditions reproducing the biological environment as closely as possible is the main goal of MD simulations. Relevant to this point, the potential energy of molecules is described by an empirical FF that is parametrized to reproduce experimental data and that represents the starting point for computing in silico the potential energy surface of the system and calculating the forces for propagating dynamic systems.

Many FFs have been developed over the last few years (Table 2). GLYCAM represents the most widely adopted FF. GLYCAM, CHARMM and GROMOS have been used to perform about 90% of the MD simulations reported since 1990 (Figure 6). The popularity of GLYCAM and CHARMM is in part due to the automation of the procedure of model parametrization.

Shifted text

In conclusion, while molecular modeling and docking provide structures of GAG-protein complexes (Figure 7A,B), the more elaborate MD simulations provide the movements of the molecules (alone or in complex) over time. This type of information is best visualized bymovies, but it can also be shown in a static way, by superimposing the structures in the most important frames (Figure 7C).

The full potential of molecular modeling, docking and MD simulations can be achieved by following the line of sequential queries schematized in Figure 8. The full set of information that can be retrieved relates not only to the 3D structure of GAGs, but also to their binding modes to targets, their binding thermodynamics and kinetics, possible allosteric effects and mechanistic insights.

## 4. Computational Studies of GAGs: What has been Done So Far

As mentioned above and summarized in Table 1, there are several reasons for the structural and functional heterogeneity of GAGs/PGs. As a result, GAG sequencing and the development of appropriate computational models have lagged behind the application of these approaches to proteins and DNA. Besides heterogeneity, other structural features of GAGs have hampered their computational modeling (Table 3). Despite all these limits, in the last ten years, we have experienced an exponential increase in the number of published papers containing GAG computational studies (Figure 9).

Interestingly, some of these papers report multiple computational studies performed on very large libraries of GAGs or GAG-mimetics, supporting the high-through put potential of computational approaches to the study of GAGs and their interaction with proteins, so important in the “omics age”. Here are some examples:(i)a whole set of MD simulations has been performed for a library of HA chains of different lengths complexed to hyaluronan lyase [92];(ii)a large array of heparin chains of different lengths has been studied in silico for their capacity to bind up to 20 different viral, animal or human proteins including sulfotransferase, heparinase, immune system-related proteins, protease inhibitors, cell adhesion proteins, blood clotting components, growth factors and their receptors [93];(iii)different GAGs (heparin and CS) of different lengths in complex with different glycosidases, chemokines, cell surface receptors and angiogenic growth factors have been subjected to computational studies [86];(iv)the in silico combinatorial library screening technology consisting of the automated construction of virtual GAGs has been employed to generate a library of heparins spanning from 2- to 8-mers that have been screened for their binding to thrombin and antithrombin [94];

To deal with the important issue of GAG length, which still represents a bottleneck and a challenge, different approaches have been described:(i)the coarse-grained modeling approach, that has been applied to a library of heparin chains spanning from 6- to 68-mers [95];(ii)dedicated algorithms have been developed to generate a library of non-sulfated chondroitin spanning from 10- to 200-mer which were compared to MD-generated ensembles for internal validation [84];(iii)the same approach was applied to libraries of HA and non-sulfated dermatan, keratan and heparan [96].

Despite some important technical progress in computational studies of GAGs (reviewed in [9] and listed in Table 2), up to 89% of GAG computational studies so far reported deal with short polysaccharide chains (from 10-mers down to 1-mer) (Figure 10).

Although the study of short GAG chains may be a deliberate choice in many instances (as in the case of pharmacologically-oriented studies of the interaction of anticoagulant heparin fragments with antithrombin), in all the other biologically oriented studies aimed at characterizing the physiological or pathological functions of GAGs/protein interactions (Figure 11), this represents a strong limit to the translation of computational predictions to biological processes, since natural GAGs can reach a length of 200 disaccharide units and GAG length is of great importance in processes such as protein homo-oligomerization [44], the formation of multimeric protein complexes [43] and cooperative binding [97], all processes that cannot be reproduced computationally and experimentally with short GAG chains.

Some other important observations can be extracted from Figure 11: about 82% of all the computational studies considered deal with the interactions of GAGs with proteins, while, of the remainder, 12% deal with GAG structures alone (see [84,95,96,98] for some examples). Among the three remaining categories, one deals with the interactions of GAGs with drugs or other inorganic or synthetic compounds (accounting for 4% of the total) [99,100,101]. Another corresponds to analyses of GAG interactions with lipids/membranes (2% of total), mainly focused on HA binding to phospholipids [102,103]. Surprisingly, we found only one computational study of a GAG–GAG interaction (namely, the anomalous interactions of HA with CS) [104].

Among the large category of GAG interactions with proteins, those with microbial proteins account for 8% of the total. Viruses are the main type of microorganism taken into consideration, with particular attention focused on those proteins exposed on the virus surface that act as determinants of infectivity by interacting with host cell HSPGs [42]. Accordingly, HS and its structural analogue, heparin, are the subjects of almost all these analyses.

Regarding human proteins, very few (less than 1% of the total) of the computational studies concern the linking of GAG chains to the core proteins of syndecan [105], glypican [106] and serglycin [107], whereas a large amount of work has been carried out for GAGs interacting with angiogenic growth factors, consistent with the great interest in the development of heparin-based antiangiogenic compounds to treat cancer [28]. Regarding the study of GAG interactions with components of the coagulation cascade, almost all of the computational analyses deal with the binding of short heparins (mainly from 4- to 8-mers) to antithrombin for the development of low-molecular weight anticoagulant heparin [108]. Worth mentioning are the computational studies of the interaction of enzymes involved in GAG metabolism, with a prevalence for degrading enzymes (e.g., heparinase/heparinase, chondroitinase, hyaluronidase) over biosynthetic enzymes (e.g., sulfotransferases), consistent with the involvement of the former in the pathogenesis of important diseases including cancer [109].

Last but not least, an important perspective from which to look at the whole body of computational studies of GAGs is their distribution with respect to the type of GAG considered (Figure 12).

Not surprisingly, almost half of the computational studies concern heparin. This is easily understandable, given the large array of biological functions played by heparin and the importance of the design of heparin-like drugs for the treatment of coagulation disorders, abnormal inflammatory or immune responses and angiogenesis-dependent diseases. Additionally, the heparin structure is more homogeneous than that of the other GAGs and it is more easily available. Thus, heparin is frequently used as a structural analogue of HS/HSPGs, both in computational and experimental studies. This has surely lowered the number of computational studies of HS, resulting in the number being significantly lower than that of studies of CS, despite the former being more biologically relevant than the latter.

## 5. Future Perspectives and Conclusions

A series of features hindered computational studies of GAGs with respect to those of other biological macromolecules (Table 3). In effect, for a long time, computational studies of GAG/protein interactions have mostly been approached in “too dry” (without modeling solvent) and/or “too rigid” (without considering structural flexibility) ways [110]. Recent advances in hardware and software technologies in this field (Table 2) have gradually allowed these neglected aspects to be included in simulations of GAG/protein interactions. Unfortunately, as in a 110-meter hurdles race, once one obstacle has been overcome, another occurs. Indeed, the increased number of parameters to be considered in GAG/protein systems has forced researchers to limit their computational studies to reasonably short GAG chains, usually no longer than 5–10 saccharide units (see Figure 10) and shorter than the significantly longer natural GAGs. This further, arduous obstacle must be overcome to unleash the full potential of computational studies of GAGs and their exploitation in the comprehension of the biological processes mediated by GAGs.

MD simulation has applications as a molecular docking-coupled technique, exploring induced fit mechanisms of GAG–protein binding, evaluating complex stability, and refining and rescoring docking poses [111]. It follows that accurate and high-throughput MD simulations of GAG–protein interactions of biological relevance require the development of suitable docking protocols with GAG models of suitable length. Unfortunately, experimental structures of long GAGs are scarce. Computational studies with long GAG chains have, however, been successful by employing GAG fragmentation, semi-flexible docking of the fragments into the binding site and subsequent chain assembly [44,81]. Although successful, these manual procedures remain laborious and time-consuming, calling for appropriate software for their automation. On this point, an automatic chain assembly method has been described that may guide the further refinement of such automated approaches for long GAG chains and their wide application in computational studies of GAGs [81].

Another remaining obstacle in the field is the absence of well-defined GAG-binding pockets on bound proteins (Table 3). In effect, dedicated computational methods to identify binding pockets in proteins have been developed that work well for small ligands [112] and short polysaccharide chains, but not for long GAG chains, whose binding is electrostatic in nature, is characterized by weak surface complementarity and is mediated by large binding surfaces (Table 3). Algorithms and software specifically dedicated to GAG–protein interactions that are able to overcome this obstacle are eagerly awaited. Databases and tools that help to evaluate GAG accessibility to proteins could speed up docking protocols. As examples, procedures involving GAG-DOCK methods [76] and electrostatic potential isosurfaces [113] have been reported that demonstrate the potential of such an approach.

In conclusion, molecular modeling, docking and MD simulation of GAGs are being actively pursued but still face challenges due to the length, flexibility and heterogeneity of GAGs. Once exploited at full potential and suitably integrated with biochemical and biological models, computational studies will contribute to a virtuous circle aimed at the deep comprehension of biological and pathological processes involving GAGs (Figure 13).

## Figures and Tables

**Figure 1 biomolecules-11-00739-f001:**
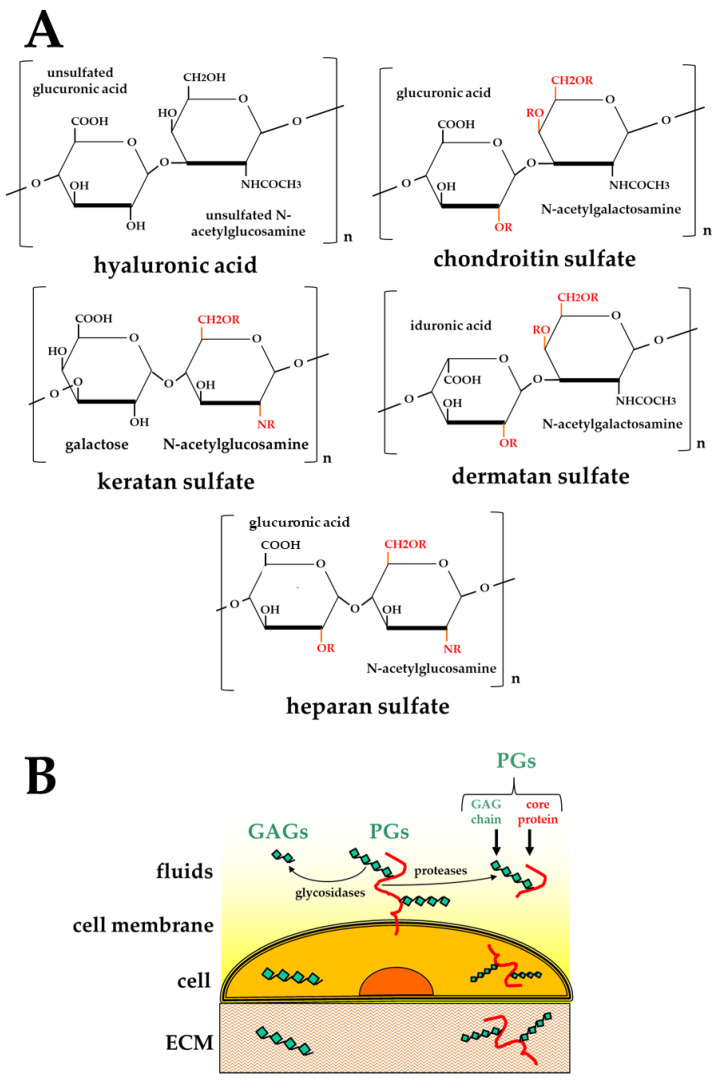
(**A**) Chemical structures of the disaccharide units composing the five main classes of GAGs. In red, “R” indicates potential points of sulfation. (**B**) Schematic representation of the distribution of GAGs/PGs inside the cell, on its surface, in the ECM or in body fluids.

**Figure 2 biomolecules-11-00739-f002:**
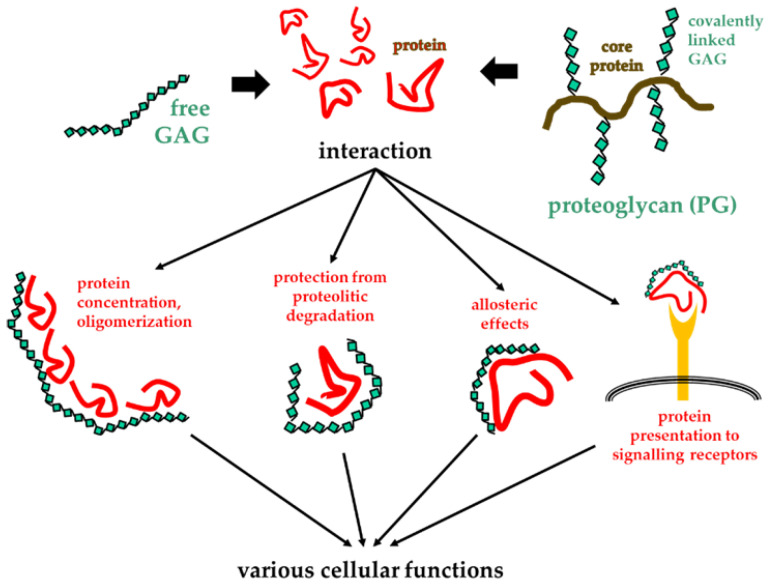
Consequences of GAG/PG interactions with proteins. Upon their binding, GAGs/PGs exert different effects on proteins that impact various cellular functions.

**Figure 3 biomolecules-11-00739-f003:**
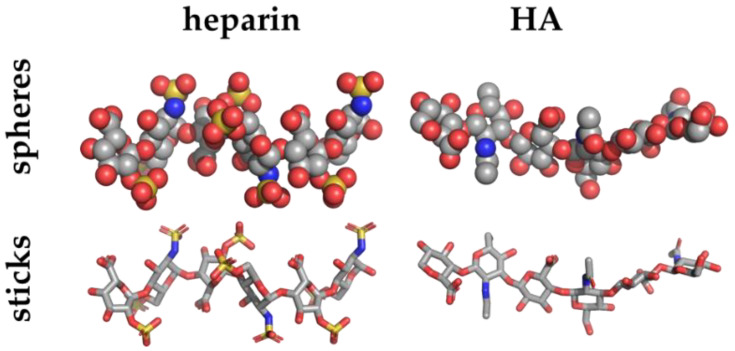
Heparin (PDBid 1HPN) [49] and HA (PDBid 1HYA) [50] crystal structures depicted in sphere and stick representations and colored by elements (carbon, oxygen, nitrogen and sulphur atoms in grey, red, blue and yellow, respectively).

**Figure 4 biomolecules-11-00739-f004:**
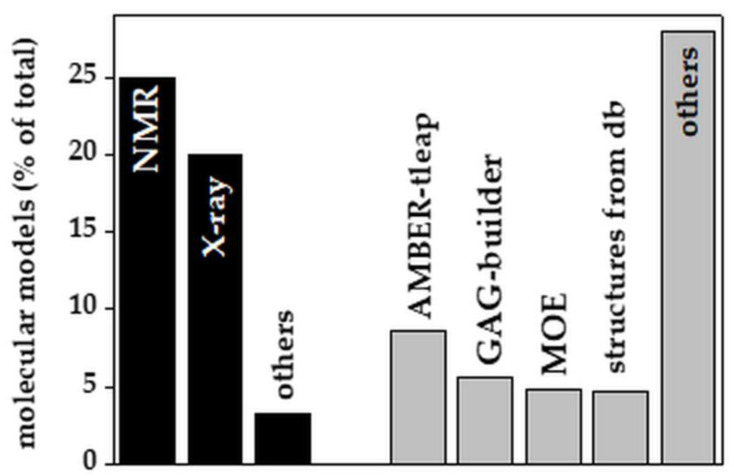
Experimental and computational methods used to generate models of GAGs alone or in complex with their binders. Each bar reports the percentage of papers in which the indicated experimental (black bars) or computational (grey bars) methods were employed. For the software grouped under “others”, see Table 2. db: database. For further details on the bibliographic research strategy, see Appendix A.

**Figure 5 biomolecules-11-00739-f005:**
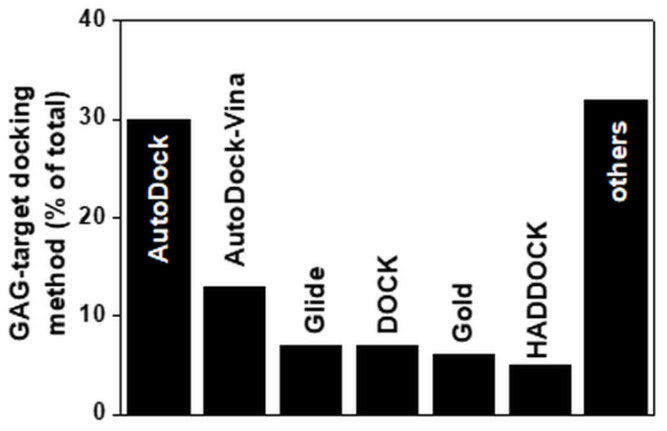
Docking software programs used to predict models of GAG complexes with their targets. Each bar reports the percentage of the published papers in which the indicated software programs were used. For the software programs grouped under “others”, see Table 2. For further details on the bibliographic research strategy, see Appendix A.

**Figure 6 biomolecules-11-00739-f006:**
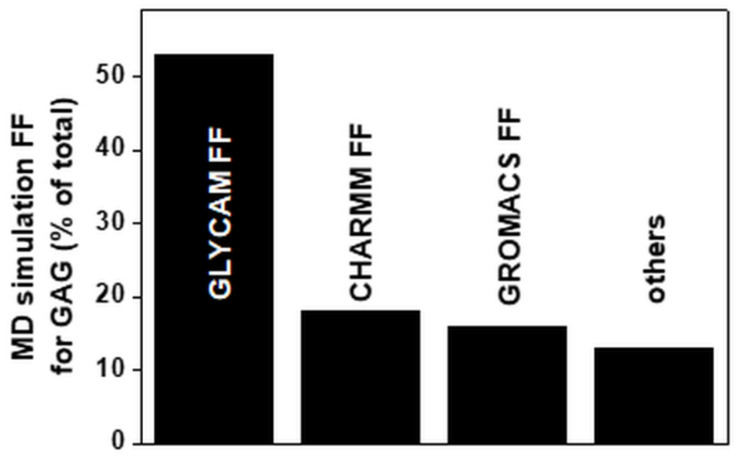
FFs used for MD simulations of GAGs alone or in complexes with targets. Each bar reports the percentage of papers in which the indicated FFs have been employed. For the FFs grouped under “others”, see Table 2. For further details on the bibliographic research strategy, see Appendix A.

**Figure 7 biomolecules-11-00739-f007:**
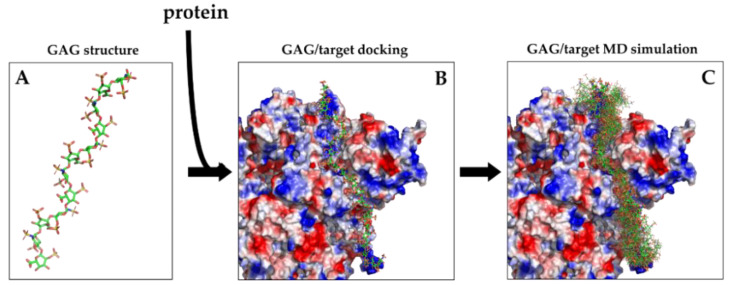
(**A**) Crystal structure of a 12-mers heparin (PDBid 1HPN) shown in stick representation colored by elements with green carbons. (**B**) Structure of a 31-mers heparin obtained with the incremental docking method [44] and docked to the spike protein of SARS-CoV2 virus shown as electrostatic potential surface to highlight the basic path to which heparin binds. (**C**) Superimposition of 20 snapshots from 1 µs of MD simulation of the 31-mers heparin/spike complex showing the cloud of conformations adopted by heparin on the protein surface (adapted from Paiardi et al. https://arxiv.org/abs/2103.07722, accessed on 12 April 2021).

**Figure 8 biomolecules-11-00739-f008:**
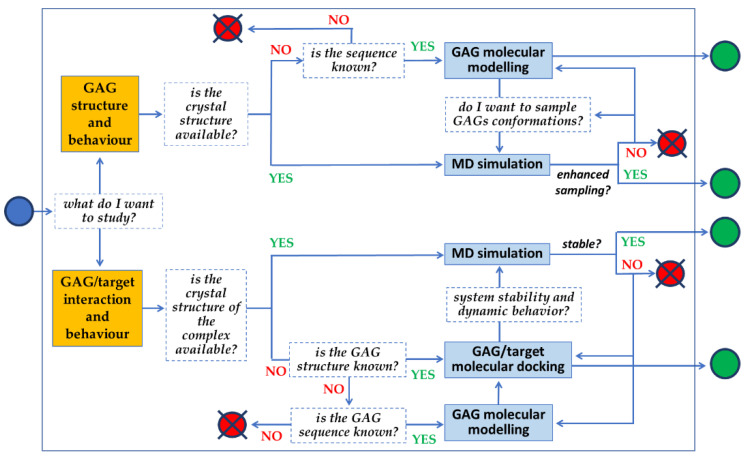
Flowchart schematizing the series of queries in an application of computational approaches aimed at a comprehensive characterization of a GAG or a GAG/target complex.

**Figure 9 biomolecules-11-00739-f009:**
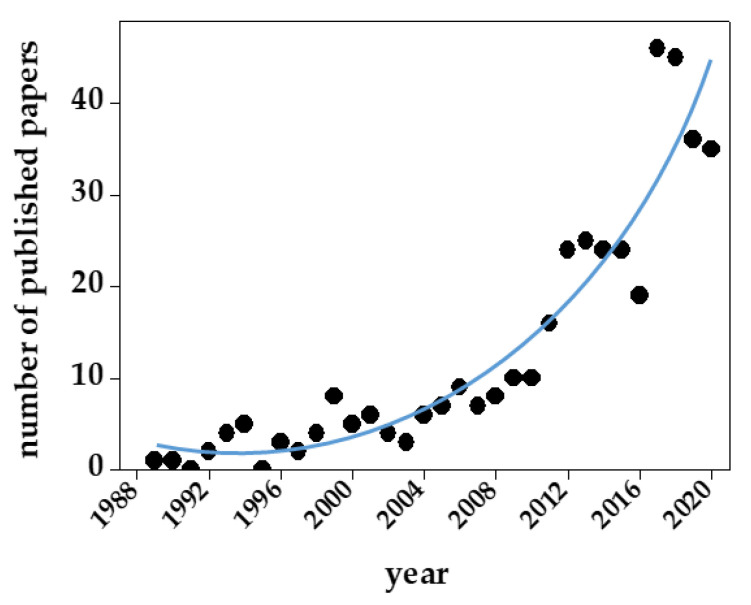
Number of papers containing computational studies of GAGs/PGs published since 1985. For further details on the bibliographic research strategy, see Appendix A.

**Figure 10 biomolecules-11-00739-f010:**
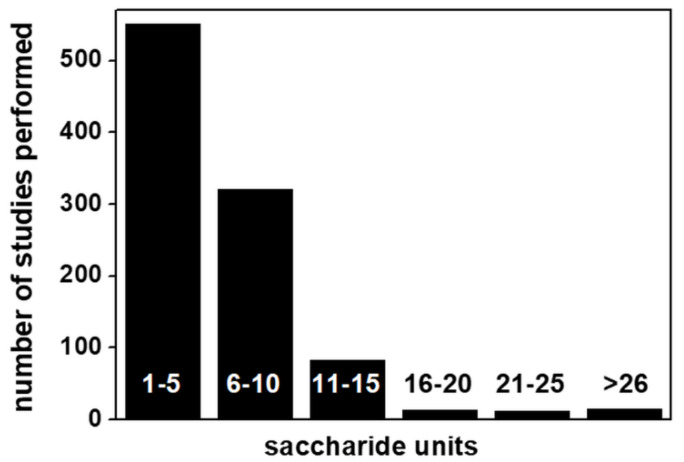
Distribution of computational studies with respect to GAG length. For further details on the bibliographic research strategy, see Appendix A.

**Figure 11 biomolecules-11-00739-f011:**
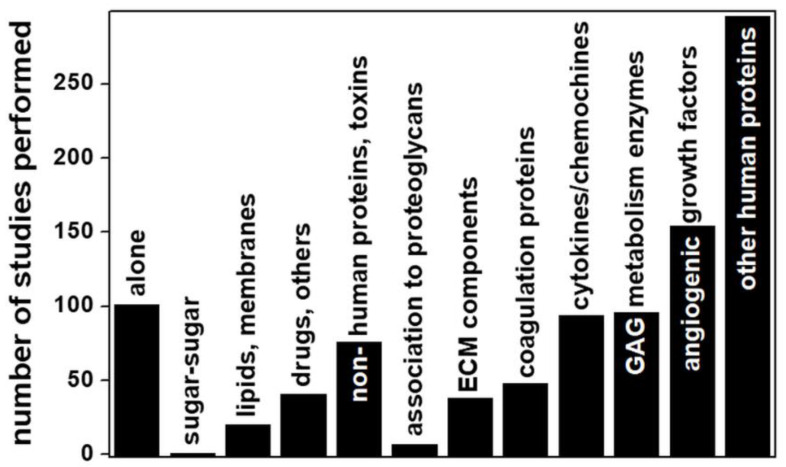
Distribution of computational structural studies of GAG alone and of GAGs complexed with the indicated ligand. For further details on the bibliographic research strategy, see Appendix A.

**Figure 12 biomolecules-11-00739-f012:**
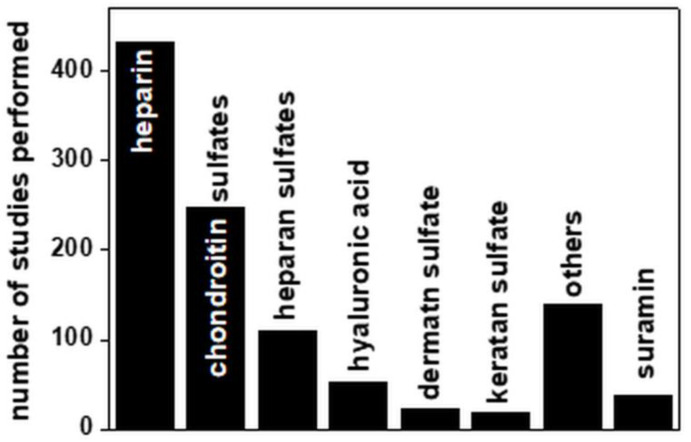
Distribution of computational studies among the different GAGs. The bar “others” includes other natural GAGs and synthetic GAG-mimicking compounds. For further details on the bibliographic research strategy, see Appendix A.

**Figure 13 biomolecules-11-00739-f013:**
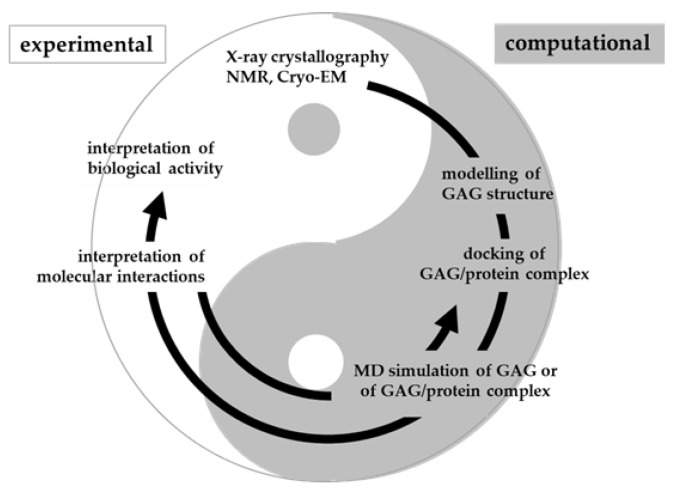
Virtuous circle between computational and experimental studies. As in the classical Yin and Yang principle, the two fields of research complement each other with the results from one pole helping the interpretation of the other. A correct balance between the two poles is needed in order to comprehend GAG/protein interactions and their biological consequences.

**Table 1 biomolecules-11-00739-t001:** List of the features that contribute to the high structural heterogeneity of GAGs/PGs.

GAGs/PGs	Combinations of hexuronic acids and amino sugars
	Length of the saccharide chain
	Positions of sulfated groups (sulfatase activity)
	Degree of sulfation (sulfatase activity)
	Distinctive expression profiles in different cell types
	Distinctive expression profiles in different tissues
	Changes of expression profile during cell differentiation
	Changes of expression profile from physiology to pathology
	Localization in intra- or extracellular compartments
	Action of different glycosidases on the GAG chain
PGs	Different core proteins
	Variable number of GAG chains attached to the core protein
	Type of association of the core protein to the cell membrane
	Action of different proteases on the core protein

**Table 2 biomolecules-11-00739-t002:** Web-based tools for computational studies of GAGs. Accession date for all the links reported in the table: 12 April 2021. FF: force field.

Name	Description (Website)	Ref.
Databases		
PDB	Bio-macromolecular structures. (http://www.rcsb.org/pdb/)	[51]
PubChem	Open chemical database containing the structures of small and large molecules including GAGs with their respective annotations (chemical structures, identifiers, physical properties, biological activities, patents, safety and toxicity data). (https://pubchem.ncbi.nlm.nih.gov)	[52]
KEGG GLYCAN	Collection of experimental GAG structures taken from CarbBank or from recent publications and present in KEGG pathways. (https://www.genome.jp/kegg/glycan/)	[53]
Zinc	Curated collection of commercially available chemical compounds in ready-to-dock, 3D formats. (https://zinc.docking.org)	[54]
DrugBank	Detailed drug properties (chemical, pharmacological and pharmaceutical features) and target information (sequences, structures and pathway). (https://go.drugbank.com)	[55]
EMBL-EBI	Collection of various tools and data from different sources (including those listed in this table) (https://www.ebi.ac.uk)	[56]
GAG-database	Comprehensive resource for 3D-structures of GAGs, oligosaccharides and their complexes with proteins (140 curated entries). (https://www.gagdb.glycopedia.eu)	[57]
monosaccharides database	Comprehensive resource for monosaccharides. (776 entries). (http://monosaccharidedb.org)	[58]
Tools to Build a GAG		
CarbBuilder	Builds GAG 3D-structures with CHARMM FF from pre-calculated glycosidic linkage torsions. (https://people.cs.uct.ac.za/~mkuttel/Downloads.html)	[59]
Chemsketch	Converts 2D drawings into 3D structures using a modified molecular mechanics approach. (https://www.acdlabs.com/resources/freware/chemsketch/)	[60]
GLYCAM-Web GAG Builder	Models GAG 3D-structures with GLYCAM06 FF using the AMBER MD package in an automated system. (http://glycam.org/gag)	[61]
CHARM-GUI Glycan Modeller	In silico N-/O-glycosylation of proteins; modeling of GAG-only systems. (http://www.charmm-gui.org/?doc=input/glycan)	[62]
Amber-tleap	Models GAG 3D-structures with the GLYCAM06 FF using the AMBER MD package. (https://ambermd.org)	[63]
MOE	Models GAG 3D-structures with MMFF94, AMBER, CHARMM FF and semi-empirical energy functions (PM3, AM1, MNDO). Conformational analysis using either a systematic or a stochastic search using random rotation of bonds. (https://www.chemcomp.com/MOE-Molecular_Modeling_and_Simulations.htm)	[64]
PRODRG	Models GAG 3D-structures with the ffgmx GROMACS FF. (http://davapc1.bioch.dundee.ac.uk/cgi-bin/prodrg)	[65]
Macromodel	Models GAG 3D-structures with MM2, MM3, AMBER, AMBER94, MMFF, MMFFs, OPLS, OPLS_2005 and OPLS3 FF. (https://www.schrodinger.com/products/macromodel)	[66]
Software for Molecular Docking		
Autodock	Stochastic local search and Lamarck genetic algorithm and empirical scoring function. (http://autodock.scripps.edu/)	[67]
Autodock-Vina	Gradient-based local search, iterated local search algorithm and empirical scoring function. (http://vina.scripps.edu/index.html)	[68]
Glide	Search algorithms include the modes of extra precision, standard precision and a high-throughput virtual filter. (https://www.schrodinger.com/products/glide)	[69]
Dock	Step-by-step geometric matching strategy; AMBER FF, empirical scoring function. (http://dock.compbio.ucsf.edu)	[70]
Gold	Genetic algorithm. (https://www.ccdc.cam.ac.uk/solution/csd-discovery/components/gold/)	[71]
HADDOCK	Encodes information from identified or predicted interfaces in ambiguous interaction restraints. (https://wenmr.science.uu.nl/haddock2.4/library)	[72]
ClusPro	Fast Fourier Transform-based algorithm and molecular mechanics energy function for scoring. (https://cluspro.bu.edu/login.php)	[73]
VinaCarb	Carbohydrate intrinsic-energy functions implemented in AutoDock Vina software. (http://glycam.org/docs/othertoolsservice/download-docs/publication-materials/vina-carb/)	[74]
GlycoTorc-Vina	Based on the VinaCarb program; uses QM-derived scoring functions to improve GAGs docking. (http://ericboittier.pythonanywhere.com/)	[75]
GAG-dock	Modification of DarwinDock method for sulfated GAGs.	[76]
FFs for GAGs		
GLYCAM_06	Set of parameters and quantum mechanical data for a collection of minimal molecular fragments and related small molecules for GAGs simulation. (http://glycam.org/docs/forcefield/)	[77]
CHARMM FF for carbohydrates	Hierarchical parametrization of model compounds containing the key atoms in GAGs. (http://www.charmm.org/charmm/resources/charm-force-fields/#charmm)	[78]
GROMOS 53A6glyc	Refined potential parameters for the determination of hexopyranose ring conformations by fitting to the corresponding quantum-mechanical profiles. (https://www.biomatsite.net/software)	[79]

**Table 3 biomolecules-11-00739-t003:** Features that makes computational docking of a GAG to a protein a challenging task.

GAGs	Long length
	Structural and chemical heterogeneity
	High flexibility
	High charge density
	Large number of torsional angles between glycosidic bonds
	Difficulty to define the impact of solvation/desolvation on GAG structure
Proteins	High charge density of GAG-binding sites
GAG/Protein Complexes	Absence of well-defined GAG-binding pockets on bound proteins
	Electrostatic nature of GAG/protein interactions
	Weak surface complementarity of GAG/protein interactions
	Indispensability of solvent for their interactions
	Impact of solvation/desolvation on GAG/protein complexes Difficulty to reproduce in silico the specific microenvironment and/or Biological setting in which GAG/protein interactions occur

## Data Availability

Data sharing not applicable.

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
