# Peer review of "A Bittersweet Computational Journey among Glycosaminoglycans"

_biomolecules, 2021, doi:10.3390/biom11050739_

Round 1
Reviewer 1 Report
This is an interesting study about” A bittersweet computational journey among glycosaminoglycans”. This manuscript described the state-of-the art of the computational approaches to the study of GAGs/PGs with the aim to point out the “bitter” and “sweet” of this field of research. This article gathered a broader understanding of linking the gap between bioinformatics and glycobiology, which have so far been kept apart by conceptual and technical differences. In addition, will provide computational scientists and glycobiologists with the fundamentals of these two fields of research, with the aim of creating opportunities for their combined exploitation, and thereby contributing to a substantial improvement of the scientific knowledge. Overall, this manuscript will attempt to bridge these two fields of research, emphasizing the opportunities for yet to be fully exploited computational approaches to the study of GAGs/PGs. The authors just need to add recent published data and references as much as possible to fit in the topic of this manuscript.
Author Response
Biomolecules-1203638
Title: A bittersweet computational journey among glycosaminoglycans
To the editor:
Please note that the modifications suggested by the reviewers required in some instance to change the layout of the manuscript.
Biomolecules-1203638
Title: A bittersweet computational journey among glycosaminoglycans
To the editor:
Please note that the modifications suggested by the reviewers required in some instance to change the layout of the manuscript. find below in bold our answer to reviewer's suggestion.
Referee # 1
This is an interesting study about” A bittersweet computational journey among glycosaminoglycans”. This manuscript described the state-of-the art of the computational approaches to the study of GAGs/PGs with the aim to point out the “bitter” and “sweet” of this field of research. This article gathered a broader understanding of linking the gap between bioinformatics and glycobiology, which have so far been kept apart by conceptual and technical differences. In addition, will provide computational scientists and glycobiologists with the fundamentals of these two fields of research, with the aim of creating opportunities for their combined exploitation, and thereby contributing to a substantial improvement of the scientific knowledge. Overall, this manuscript will attempt to bridge these two fields of research, emphasizing the opportunities for yet to be fully exploited computational approaches to the study of GAGs/PGs. The authors just need to add recent published data and references as much as possible to fit in the topic of this manuscript.
This has been done now (see references 5, 6, 14, 23, 24, 32, 36, 37, 80).
Reviewer 2 Report
Biomolecules-1203638
Title: A bittersweet computational journey among glycosaminoglycans
The manuscript by Paiardi et al. summarizes the current knowledge on computational approaches to study glycosaminoglycans and proteoglycans at structural and functional levels. Overall, the work is very interesting and of high relevance since the computational approaches applied to GAGs still remain little explored in comparison with other biological molecules. The review is clearly written and includes updated and appropriate references. It contains information that may be shared by the bioinformatics and glycobiology communities and bridge these two fields of research.
I have a few suggestions to improve the manuscript:
Major comments:
- Taking in consideration that several figures of the manuscript are graphs showing the number of publications/papers that apply specific methods/ software, number of computational studies, etc, the reviewer considers that it would be important to include in the manuscript a detailed description of how these numbers were calculated. This description should include what search strategy was applied by the authors (indicating selection criteria such as reference keywords), what was the interval of time considered for this study (from1990 to…) and the publication databases that were used.
Minor comments:
- Lines 62 and 63: Regarding the reference to “glycosaminoglycomics” analyses it would be relevant to include and refer the following reference: Chen YH, et al The GAGOme: a cell-based library of displayed glycosaminoglycans. Nat Methods. 2018 Nov;15(11):881-888. doi: 10.1038/s41592-018-0086-z.
- Figure 1A, the glucuronic acid representation in HS structure needs to be corrected (in the present form it is represented as iduronic acid);
- Lines 150-155: when referring to GAG biosynthetic pathways, and particularly to HS synthesis, it could be valuable to add the following recent review:
Annaval T, el al. Heparan Sulfate Proteoglycans Biosynthesis and Post Synthesis Mechanisms Combine Few Enzymes and Few Core Proteins to Generate Extensive Structural and Functional Diversity. Molecules. 2020 Sep 14;25(18):4215. doi: 10.3390/molecules25184215.
- Table 1, it would be relevant to include in the Table 1 and also briefly mention in the text the contribution of sulfatases activity for the structural heterogeneity of GAGs/PGs.
- Line 194, in the text Table 3 is cited before Table 2, this should be corrected.
- Lines 215-220: when referring the aberrant expression of GAGs and PGs in cancer it would be relevant to cite two very recent reviews on this topic:
Faria-Ramos I, eta la. Heparan Sulfate Glycosaminoglycans: (Un)Expected Allies in Cancer Clinical Management. Biomolecules. 2021 Jan 21;11(2):136. doi: 10.3390/biom11020136.
Hassan N, Greve B, Espinoza-Sánchez NA, Götte M. Cell-surface heparan sulfate proteoglycans as multifunctional integrators of signaling in cancer. Cell Signal. 2021 Jan;77:109822. doi: 10.1016/j.cellsig.2020.109822.
- Table 3 line 6, start sentence with capital letter.
- Lines 397-398: the following sentence needs rephrasing: “Other obstacles are the flexibility of the functional groups on the monosaccharides the flexibility of the whole GAG chain…”
- Lines 448-449, the sentence needs rephrasing to become more clear.
Author Response
Biomolecules-1203638
Title: A bittersweet computational journey among glycosaminoglycans
To the editor:
Please note that the modifications suggested by the reviewers required in some instance to change the layout of the manuscript. Find here below in bold our answers to reviewer's comment ans suggestions.
Referee # 2
The manuscript by Paiardi et al. summarizes the current knowledge on computational approaches to study glycosaminoglycans and proteoglycans at structural and functional levels. Overall, the work is very interesting and of high relevance since the computational approaches applied to GAGs still remain little explored in comparison with other biological molecules. The review is clearly written and includes updated and appropriate references. It contains information that may be shared by the bioinformatics and glycobiology communities and bridge these two fields of research.
I have a few suggestions to improve the manuscript:
Major comments:
- Taking in consideration that several figures of the manuscript are graphs showing the number of publications/papers that apply specific methods/ software, number of computational studies, etc, the reviewer considers that it would be important to include in the manuscript a detailed description of how these numbers were calculated. This description should include what search strategy was applied by the authors (indicating selection criteria such as reference keywords), what was the interval of time considered for this study (from1990 to…) and the publication databases that were used.
We have now added appendix A (second pg. 20) in which we describe the publication databases used, reference keywords and interval of time considered. As a consequence, legends to Figures 4-6 and 9-11 have changed, also describing how the values were calculated and expressed.
Minor comments:
- Lines 62 and 63: Regarding the reference to “glycosaminoglycomics” analyses it would be relevant to include and refer the following reference: Chen YH, et al The GAGOme: a cell-based library of displayed glycosaminoglycans. Nat Methods. 2018 Nov;15(11):881-888. doi: 10.1038/s41592-018-0086-z.
This has been now done
- Figure 1A, the glucuronic acid representation in HS structure needs to be corrected (in the present form it is represented as iduronic acid);
We wish to thank the reviewer for pointing out this error. The structure has been now corrected
- Lines 150-155: when referring to GAG biosynthetic pathways, and particularly to HS synthesis, it could be valuable to add the following recent review: Annaval T, el al. Heparan Sulfate Proteoglycans Biosynthesis and Post Synthesis Mechanisms Combine Few Enzymes and Few Core Proteins to Generate Extensive Structural and Functional Diversity. Molecules. 2020 Sep 14;25(18):4215. doi: 10.3390/molecules25184215.
We have now replaced the previous reference (dated 2002 with the more recent one suggested by the reviewer.
- Table 1, it would be relevant to include in the Table 1 and also briefly mention in the text the contribution of sulfatases activity for the structural heterogeneity of GAGs/PGs.
Following reviewer’s suggestion we have now included the role of sulfatases activity in table 1 and in the following text paragraph.
- Line 194, in the text Table 3 is cited before Table 2, this should be corrected.
Again, we wish to thank the reviewer for pointing out this error. The indicated Table 3 is indeed Table 1. This has been now corrected.
- Lines 215-220: when referring the aberrant expression of GAGs and PGs in cancer it would be relevant to cite two very recent reviews on this topic: Faria-Ramos I, eta la. Heparan Sulfate Glycosaminoglycans: (Un)Expected Allies in Cancer Clinical Management. Biomolecules. 2021 Jan 21;11(2):136. doi: 10.3390/biom11020136. Hassan N, Greve B, Espinoza-Sánchez NA, Götte M. Cell-surface heparan sulfate proteoglycans as multifunctional integrators of signaling in cancer. Cell Signal. 2021 Jan;77:109822. doi: 10.1016/j.cellsig.2020.109822.
The references suggested by the reviewer have been now included
- Table 3 line 6, start sentence with capital letter.
This has been now done
- Lines 397-398: the following sentence needs rephrasing: “Other obstacles are the flexibility of the functional groups on the monosaccharides the flexibility of the whole GAG chain…”
The sentence has now been rephrased.
- Lines 448-449, the sentence needs rephrasing to become more clear.
Also this sentence has been rephrased.